# Solar Panel Detection within Complex Backgrounds Using Thermal Images Acquired by UAVs

**DOI:** 10.3390/s20216219

**Published:** 2020-10-31

**Authors:** Jhon Jairo Vega Díaz, Michiel Vlaminck, Dionysios Lefkaditis, Sergio Alejandro Orjuela Vargas, Hiep Luong

**Affiliations:** 1Centro de Investigación en Ciencias Básicas y Aplicadas, Doctorado en Ciencia Aplicada, Universidad Antonio Nariño (UAN), 110231 Bogotá, Colombia; 2Telecommunications and Information Processing (TELIN), Ghent University, Imec, B-9000 Ghent, Belgium; michiel.vlaminck@ugent.be (M.V.); hiep.luong@ugent.be (H.L.); 3Sitemark, Gaston Geenslaan 11, 3001 Leuven , Belgium; dionysios.lefkaditis@sitemark.com; 4Facultad de Ingeniería Mecánica Electrónica y Biomedica, Universidad Antonio Nariño (UAN), 730001 Ibagué, Tolima, Colombia; seorjuela@uan.edu.co

**Keywords:** solar panel detection, solar panel projection, texture descriptor, support vector machine, deep learning, NIR, thermal imaging

## Abstract

The installation of solar plants everywhere in the world increases year by year. Automated diagnostic methods are needed to inspect the solar plants and to identify anomalies within these photovoltaic panels. The inspection is usually carried out by unmanned aerial vehicles (UAVs) using thermal imaging sensors. The first step in the whole process is to detect the solar panels in those images. However, standard image processing techniques fail in case of low-contrast images or images with complex backgrounds. Moreover, the shades of power lines or structures similar to solar panels impede the automated detection process. In this research, two self-developed methods are compared for the detection of panels in this context, one based on classical techniques and another one based on deep learning, both with a common post-processing step. The first method is based on edge detection and classification, in contrast to the second method is based on training a region based convolutional neural networks to identify a panel. The first method corrects for the low contrast of the thermal image using several preprocessing techniques. Subsequently, edge detection, segmentation and segment classification are applied. The latter is done using a support vector machine trained with an optimized texture descriptor vector. The second method is based on deep learning trained with images that have been subjected to three different pre-processing operations. The postprocessing use the detected panels to infer the location of panels that were not detected. This step selects contours from detected panels based on the panel area and the angle of rotation. Then new panels are determined by the extrapolation of these contours. The panels in 100 random images taken from eleven UAV flights over three solar plants are labeled and used to evaluate the detection methods. The metrics for the new method based on classical techniques reaches a precision of 0.997, a recall of 0.970 and a $F_{1}$ score of 0.983. The metrics for the method of deep learning reaches a precision of 0.996, a recall of 0.981 and a $F_{1}$ score of 0.989. The two panel detection methods are highly effective in the presence of complex backgrounds.

## 1. Introduction

The increased use of renewable and low-carbon energy has led to economic [1] and environmental benefits [2]. Among the renewable sources is the use of solar energy in for example on the rooftop of houses [2,3], buildings [4] or on wide fields for power plants [5]. Solar energy is commonly captured by photovoltaic panels. However, the efficiency of the panels is deteriorates in the presence of anomalies, such as hot spots. Hot spots occur due to non-uniform energy generation between the photo-voltaic cells, with differences that exceed 5% of the temperature admitted in standard test periods. These sudden temperature rises have the potential to lead to spontaneous ignition [6].

Anomalies in solar panels lead to energy and temperature changes, so they are measured with current and voltage indicators [7,8] and thermal sensors [9,10,11]. The anomalies measured by thermal sensors are changes in energy efficiency [12], material fatigue [13,14] and hot spots [15,16]. However, hot spots can be catalogued at different levels of failure, which are associated with their geometry [17]. Since the classification of anomalies with thermal cameras is based on their geometry, a first step for their correct classification is the identification of the solar panel. Related research has also focused on the detection of the solar panels array [18,19].

Due to the creation of large solar plants, it has been required to incorporate the use of drones for the inspection of massive amounts of solar panels [20]. The images generated by this technology, when processed with photogrammetric methods, allow generating orthomosaics and incorporating thermal and RGB image layers to geographic information systems (GIS) [21,22,23]. For the identification of the panels, information from the thermal image has been used, together with the RGB image [24] or the 3D models [25]. However, the developments are focused on the identification of the panels using only thermal video [26,27] or thermal imaging [28].

Panel detection focuses on identifying rectangular structures. However, this identification is difficult in thermal images because not all panel edges are visible [24] and because of irregularities by weeds shades, sunlight reflection [20] or hot spots. By adding this type of diagnostic difficulties with other variables such as different flight heights, changes in lighting [29], panel like structures, energy lines and images with lens distortion, which are all considered as complex backgrounds [30] (Figure 1).

The identification of solar panels in thermal images with complex backgrounds has five challenges:Hot spots create an atypical distribution of data, which leads to a loss of image contrast.The edges suffer from distortion and diffusion.There are structures that have a panel-like geometry.Edge detection fails due to image saturation in areas with hot spots or sunlight reflections.Other occlude obstruct the panel, making its full geometry invisible.

This paper focuses on the first step for a proper diagnosis of the fault, in which a solar panel detection is necessary to delimit the region of interest for the detection and classification of the anomaly, because the correct classification is based on geometries in the panel or between panels. The development of methodologies for the detection of solar panels directly from the thermal image will allow the design of a fully automated anomaly classification system with real time processing. Therefore, this work is focused on reducing panel detection failures, since those that are not detected are those that typically contain anomalies. So, the main contributions are:A novel method based on classical Machine Learning is developed for the detection of solar panels using effective preprocessing steps.our novel method based on classical Machine Learning approach is compared against a state-of-the-art Deep Learning algorithm.A novel method predicting missing (i.e., undetected) solar panels. This method is used as a post-processing step for both detection algorithms: based on classical techniques and deep learning. The post-processing improves the recall of both methods.

### Related Work

From the investigations with images taken with drones that only use thermal images as reference, some researches only focused on detecting the damage without discriminating the panels. To identify the damage, they calculate the temperature difference [31] with Canny edge detection [26]. However, the presence of hot spots creates an atypical distribution of data, which leads to a loss of image contrast, making it difficult to identify the panels. Alternatively, the panels were segmented by the following methods: Template Matching [32], segmenting the image using k-means clustering [33], the active contour level sets method (MCA) and the area filtering (AF) approach [28]. These methods have been referred to as classical methods, which fail to focus only on identifying panels and do not include a method for differentiating them from the environment with panel-like structures.

For the detection of the panels among the models used in deep learning are: an algorithm based on a fully convolutional neural network and a dense conditional random field [30] and a convolutional neural network framework called ’You Only Look Once’ (YOLO) [29]. However, the presence of energy lines creates atypical geometries that none of the methods can overcome.

The weaknesses of the related works are grouped into three categories: the first category focuses only on the detection of hot spots; the second category uses photos with panels in high resolution and do not reference solutions for the features of the complex backgrounds; the third category handles deep learning techniques, but does not propose methods for projecting the missing panels. This information is presented in more detail in Table 1.

## 2. Materials and Methods

### 2.1. Experimental Setup and Study Area

The thermal images were acquired with a DJI Zenmuse XT camera developed by FLIR [36], with 640 × 520 pixels, in TIF format with 14 bits. The data set contains 11 flights from three different solar plants, with a relative flight height of 34 to 56 m. The images acquired from these solar plants presented have complex backgrounds, and all the analysis is based only on the information available in the thermal image, so, other flight plan information is not included. In total, 18,244 panels are delimited out of 100 randomly selected thermal images.

Image processing was coded in the python 3 language with CUDA toolkit, the principal libraries are NumPy [37], OpenCV [38], scikit-image [39], SimpleITK, PyRadiomics [40,41] and Detectron 2 [42].

The new method based on classical techniques is described in Section 2.2. The method based on deep learning is described in Section 2.3. These methods are evaluated with the precision metrics described in Section 2.4.

### 2.2. Solar Panel Detection Using Our New Method Based on Classical Techniques

The first method to detect solar panels consists of the following steps: first an image correction; second, an image segmentation; third, a segment classification with machine learning; finally, a post-processing step based on the detected panels (Figure 2). The proposed method highlights the use of the OpenCV library, since it allows to calculate a contour and its characteristics such as: the image area, a rotated rectangle, the angle of rotation and convert the contour into a segment of the image, among others [38].

#### 2.2.1. Image Correction

In image processing some data normalization steps are often required. However, in cases where hot spots or areas with sun reflection occur, the normalization implies the generation of images with low contrast and loss in the textural characteristics. Therefore, to avoid the loss of subtleties in the image it is proposed to remove the outlier data in the following way: the average plus twice the standard deviation is used as the reference value. Outliers are corrected if less than 5 percent of the data exceeds the reference value. To reduce the distribution tail, the outliers are assigned to the reference value (See Figure 3).

#### 2.2.2. Image Segmentation

The detection of panels is based on the detection of their edges, which have a temperature lower than the center of the panel. Among the basic concepts used is that panels are four-sided structures that can be simplified as rectangles. However, solar panels on complex backgrounds have anomalies that lead to distortions, loss of continuity at the edges and diffuse edges that make it difficult to detect the rectangles. Therefore, the proposed algorithm consists of an image preprocessing, edge detection and segment determination. The image preprocessing uses a convolution filter, a bilateral filter and a gamma correction. The edge detection involves threshold settings at different ranges in order to determine most of the image edges. Each edge detection process involves the calculation of contours that are evaluated in shape and size. These contours are drawn as a mask in an iterative way in an image, which at the end collects all the segmentations made.

Correct line detection is based on changes in the pixel values that identify a border. However, the edges of certain panels are represented only by a dashed line one pixel wide and therefore our method seeks to make these types of edges visible. The image preprocessing has the following steps: first, a convolution filter is used with the following mask [1 1 1; 1 1 1; 1 1 1]. Usually, this filter is used in other contexts to generate noise, but in this case it is used to increase the width of the edge due to the displacement of the data by one pixel around it. Second, in certain panels in their corners there are no marked differences in temperature with their surroundings and in other cases the edges are diffuse, so a bilateral filter is used to homogenize the data in 3-pixel windows with the aim of emphasizing the borders. So, the bilateral filter is applied with the following configuration: 3 range pixels, a sigma filter for the color space of 200 and a sigma filter in space coordinates of 50. With this configuration, the edges are preserved by having a greater variation in intensity between the pixels. Third, In order to emphasize the edges of the panels that do not contrast with their surroundings, the contrast of the image is enhanced by using a gamma correction with a very high value such as 4.8 (Figure 4a). The value was selected after evaluating different settings on images with low edge contrast.

The temperature of the panels edges vary locally. Therefore, the edge detection uses an adaptive threshold, but the widths of the panel edges are of variable size. Therefore, our algorithm is used with three windows of different width (5, 11 and 21) so that with each pixel window panels are detected that with the other one are not (Figure 4b). After thresholding each of the images, edge detection is performed, calculating the internal contours of the panels. Since a panel is a rectangular structure, this step seeks to determine whether a segment contain that characteristic. The segment determination consists of a classification of the contours by their image area, the number of corners and their solidity. The contours are selected if they have: an area between 100 square pixels and the expected panel size, three to four corners and a solidity greater than 0.8. The contour is used to determine an area of the image called a segment, It is assumed that an segment smaller than 100 square pixels does not represent a panel. The number of contour corners are counted from the approximation of the contour to a shape with an OpenCV function using the Ramer–Douglas–Peucker algorithm. “Solidity is the ratio of contour area to its convex hull area” [38]. Finally, The segments are eroded to avoid overlapping and these are drawn as a mask on a zero matrix shaped like the original image (Figure 4c). To facilitate the evaluation each segment is drawn with a different color on the grayscale corrected image (Figure 4d).

#### 2.2.3. Panel Classification with Svm

This is one of the most important steps of the proposed algorithm, because up to this point there is a series of contours that has a similar shape to a rectangle, but it is not known if this is a panel or not, so the method uses a supervised classification process based on the texture of the panels.

Using the mask with rectangular segments, the contours are calculated again. Each contour is converted to a segment of the image and this is dilated by circular structure element with a 6 pixel diameter to include the panel edges. An optimized set of texture descriptors are calculated for each segment, and it is classified whether it corresponds to a panel or not using a support vector machine (SVM).

The texture descriptors are calculated with the PyRadiomics library. This is a library developed for the extraction of features from medical images, which are based on texture descriptors. However, this library was chosen because it implements different transformations in the original image and characterizes it with different texture descriptors, allowing to evaluate subtle changes of the image. Moreover, with a configuration for 2D images it can be used in other contexts. The algorithm requires an image and a mask of the feature extraction area. The images must be converted to a Insight ToolKit image (ITK) [43] with the SimpleITK library. Two elements are parameterized in order to calculate as many features as possible. The algorithms that characterize the image with the specific texture descriptors (FeatureClass) and the image type (ImageType) which is the original image and its transformations. Six algorithms for texture decriptors (shape2D, firstorder, Gray Level Co-occurrence Matrix (GLCM), Gray Level Run Length Matrix (GLRLM), Gray Level Size Zone Matrix (GLSZM) and Gray Level Dependence Matrix (GLDM)) are applied on five transformed images (Original, LoG, sigma [3.0, 5.0] and Wavelet). This results in a total of 440 texture descriptors [40]. In this case the use of the image generated by the local and binary pattern algorithm was excluded because it was not efficient for classification.

A support vector machine is then used for the classification of a texture descriptor vector. Due to the amount of data, it uses the implementation of a regularized linear model with stochastic gradient descent (SGD) learning. The ten-fold cross validation method is used to determine the classification accuracy. After evaluating different configurations, a larger classification efficiency is obtained by: standardization of features by removing the mean and scaling to unit variance, max_iter = 1000, class_weight = “balanced”, learning_rate = “adaptive”, eta0 = 0.5, penalty = “elasticnet”, and tol = 1 × 10${}^{3}$ [39,44].

The optimized texture descriptor group is calculated from a created database of texture descriptors with data corresponding to two classes: solar panels and other objects. The descriptors with a probability of less than 0.001 in an ANOVA test and a classification accuracy with SVM of more than 90 percent in cross validation are selected. Finally, a mask is created with the selected contours corresponding to the solar panels of the thermal image. To facilitate the evaluation each segment is drawn with a different color on the grayscale image (Figure 5).

#### 2.2.4. Post-Processing

A final step is proposed in which the detected panels are used to infer the location of the panels that possibly remained undetected. This algorithm includes three steps: first, the selection of contours by image area and angle of rotation; second, the projection of edges of each rotated rectangle; third, the determination of new contours and their classification. The last two steps are repeated until no new contours are selected.

The selection of the contours of the detected panels is based on the following premises: the solar panels have four sides, the edges of the image have cut-out panels and most of the contours are solar panels. Therefore, each contour can be simplified to a rotated rectangle and the coordinates of the image where there are incomplete panels are related to the coordinates of the rotated rectangle. The rotated rectangle is composed of four points in the image, these points are represented by X and Y coordinates. The maximum X and Y shift value between points of each rotated rectangle is calculated and stored in a matrix. Finally, the section of the image border with incomplete panels, is calculated from the maximum value of the matrix of displacements in X and Y, plus a 20%. It is assumed that the region where there are incomplete panels is a rectangle with an edge shift towards the center with the maximum X value for vertical edges and the maximum Y value for horizontal edges. Thus, excluding the panels at the edges, the average area and the angle of rotation of the rotated rectangles are calculated. For the panels at the edges only the angle of rotation of the contour is calculated. Because images suffer from lens and perspective distortions, the contours with a 20% deviation from the rotation angle and a deviation of 20% below and 30% above the average area are eliminated (Figure 6).

The projection of the edges of the rotated rectangle has as a premise that the solar panels are close to others in a structure that forms a grid. Therefore, the panels are represented by a rotated rectangle and each rectangle has four sides. Each side is two points joined together to represent a line, which is projected 1.1 times towards its ends. The projected lines of all rotated rectangles are drawn in a zeros image (Figure 7).

The determination of new contours and their classification is based on the premise that the panels have a homogeneous shape, are grouped and can be classified by texture descriptors. So, the contours are calculated from the matrix of zeros with the lines drawn. The contours are classified in different steps: first the contours with an area deviating less than 30% from the average area are selected. Second, contours that do not overlap with the detected panels are selected. Third, they are classified according to whether they belong to the panel structure or not. If they cannot be associated with the structure, they are classified using a support vector machine (see Section 2.2.3) (Figure 7).

To evaluate whether an outline belongs to a panel structure, the following assumptions are made: if the detected panels are close enough, they are in structures and the pixels can be classified into two classes, one of which represents the panel structures. The first premise is represented in a mask, which is created in the following way: First, all detected panels are expanded until they overlap with their nearest neighbour. Second, all panels are drawn in a mask. Third, the contours of the mask are calculated. Fourth, the convex hull is calculated for each contour. Fifth, a mask is drawn with the convex hulls. The second premise is taken to a mask with the following steps: First, a bilateral filter is applied to the corrected image. Second, the pixels are classified in two classes using K-means. Third, a mask is drawn with the classification. Fourth, a morphological function of opening and closing is applied to eliminate the noise. Fifth, the label corresponding to the detected panels is determined. The selection takes into account the contour belonging to the convex hull mask and the k-means mask. Since undetected panels in the corners do not intersect with the entire area of the convex hull mask, then at least a 20% overlap is accepted. Because the k-means mask can have classification errors, it is acceptable for the contour to have 85% of the pixels classified as panel area.

### 2.3. Panel Prediction with Deep Learning

#### 2.3.1. Detectron2

Regarding the panel prediction using deep learning, the PyTorch-based modular object detection library Detectron2 is employed. Detectron2 also has implementation of mask R-CNN and mask R-CNN have greater accuracy than those based on R-CNN, fast R-CNN, YOLO among others [45]. Furthermore, The Detectron2 library was chosen because it has better results on benchmarks compared to other popular open source mask R-CNN implementations [42]. The Mask R-CNN network is trained with thermal images with only one class: solar panels. With the support of the algorithm described in Section 2.2 and a graphical interface, all the panels present in an image are labelled in COCO format, even the panels at the edges of the image. Labeling is done in the visual interface by a user who eliminates false positives and corrects false negatives. Because Detectron2 only supports 8-bit images, a pre-treatment of the image is done and it is converted to three channels.

To accentuate the edges, a three-channel image is created as follows: first, a sigmoidal normalization of the 14-bit TIF image is applied; then the data is transformed in the range 0 to 255; the first channel has no transformation; the second channel has a bilateral filter as described in Section 2.2.2; and the third channel has a gamma correction as described in Section 2.2.2.

The detectron2 model is trained with the procedures reported for a new data set [42], the model is mask_rcnn_R_50_FPN_3x. The Mask R-CNN network is trained using 2000 cycles and a custom data loader. The custom data loader is used for data augmentation, with random changes of: 1 percent rotation, horizontal flip, contrast between 0.5 and 1.5, brightness between 0.6 and 1.5, lighting of 2, saturation between 0.5 and 1.5 and a crop function. Through previous evaluations it was determined that the prediction with the score threshold of 0.2 does not affect the false positives and allows not to exclude panels of difficult detection. Finally, the results are drawn on an image and each prediction is exported to a mask in a NumPy file [37].

#### 2.3.2. Post-Processing

This algorithm follows the same process of Section 2.2.4 except for the following variation in the first step where the contours of the detected panels are selected. First, the predictions of the panels are drawn in masks and from these the contours are calculated and eroded to avoid overlapping. Second, before calculating the average area, the data is debugged by removing the tails of the distribution. Third, the rotation angle of the rotated rectangle is not used for the contour selection.

### 2.4. Evaluation

The detection efficiency of the methods is evaluated by a test data set, which is different from the training data set. The novel method based on classical techniques consists of a training part for the texture descriptors. The texture database is not part of the images selected for evaluation, so all labeled images are used as a test image. Prediction with deep learning requires training, therefore, to include all labeled images in the test the concept of cross validation is used. The images were divided into five groups, so that in each group 80% is used for training and 20% for testing, and the images selected for testing are different for all groups (Figure 8).

All panel detection steps of both methods are evaluated with precision metrics (Figure 8), the classification metrics of a single class are used, in which positive and negative values are taken into account [46]. The metrics used to evaluate the effectiveness of classification are precision, recall and F-measure [32].

The precision represents the proportion of positive samples that were correctly classified compared to the total number of predicted positive samples [47] as indicated in Equation (Equation 1), where tp indicates the number of true positives and fp indicates the number of false positives:(1)$$Precision=\frac{tp}{tp+fp}$$

The recall of a classifier represents the correctly classified positive samples to the total number of positive samples [47] and it is estimated according to Equation (Equation 2), where fn indicates the number of false negatives:(2)$$Recall=\frac{tp}{tp+fn}$$

The main F-measure is the $F_{1}$ score which is defined as the harmonic mean of precision and recall [48]. It is estimated according to Equation (Equation 3):(3)$$F_{1}=2\cdot \frac{Precision\times Recall}{Precision+Recall}.$$

The detected panels are drawn on the image and through visual inspection the false positives are determined. We define a false positive as a selection that is not a panel or a panel that presents: center shift, associate more than one panel, or an area less than 90%. The false negatives are the undetected panels. The total number of panels is the total number of labels used for training. The value of the true positive is the total number of panels minus the false negatives.

## 3. Results

Regarding the panel classification described in Section 2.2.3, 440 texture descriptors are calculated and a vector of 290 texture descriptors is optimized. A database with 1555 panel vectors and 302 non-panel vectors is created by a visual classification. This database is used to train a support vector machine. The database has a classification efficiency of 91% as it was evaluated in 10-fold cross validation.

For the deep learning based method, predictions of images not used for training were made. To that end the thermal photos were divided into five groups and 80 photos were used for training and 20 photos for prediction, alternating the groups so that the prediction photos were not repeated.

The detections made by each method in each step were drawn on the original image. These images were used for visual inspection and determining the false negatives and false positives.

Figure 9 shows five cases of thermal images with complex backgrounds and panel detection with a novel method based on classic techniques with a post-processing step. The first is an image with sun reflection; the second is an image with the presence of overhead power lines with cuts close to 45 degrees with the geometry of the panels; the third an image with the presence of energy lines with cuts close to 90 degrees with the geometry of the panel; the fourth is an image with poorly defined panel edges; and the fifth presents structures like panels. In these images they show the detections made in each step of this method. Therefore, in the first step (see Section 2.2.2) the images show numerous false positives, which are reduced with the second step (see Section 2.2.3) when sorting with a support vector machine. Ending with the drawing of panels that have not been detected (see Section 2.2.4). However, due to the selection of outlines in the last step, some distorted panels (due to the distortion of the image by the camera lens) are removed from the edges.

The images of Figure 9 are consistent with the data shown in Table 2. This table shows that the classical method in complex environments without machine learning has a precision value of 0.886, which would lead to dismissing this method. However, the addition of the post-processing step allows to improve the quality of the method raising the value of precision to 0.997. However, the number of panels detected does not have such a large rise as the recall value is only increased by 0.009. This is because in the post-processing step some detections are eliminated because they are distorted. These variations are reflected in the value of the $F_{1}$ score, which has an increase between the first and third step of 0.06.

Figure 10 shows the same five cases of thermal images with complex backgrounds presented in the Figure 9 and panel detection with deep learning and a post-processing step. These images show that the deep learning method leads to a greater stability in the shape of the panel, with the four sides well defined. However, this method has failures in images with power lines, poorly defined panel edges and panel-like structures. These detection failures are partially solved in the post-processing step.

Table 3 shows the values of the precision metrics for the test evaluations in the deep learning prediction and drawing step. The precision value shows that the deep learning method has few errors, but the recall shows the sensitivity of the method to the training data, so training with a large database is recommended. In this table, it is shown how the post-processing step affects the accuracy metrics. This is demonstrated in the change of the total data of the groups, reducing the false negatives by more than 60% that lead to an increase of the recall value of 0.032, reaching a precision value of 0.996.

## 4. Discussion

The proposed methods solve the problems related to the complex backgrounds mentioned in this article. For the new method based on classical techniques, the following points were verified: first, the correction posed in Section 2.2.1 was effective in the case of images with an atypical distribution of data due to hot spots. Second, that the procedure posed in Section 2.2 allows the accentuation of diffuse edges. Third, that the classification with support vector machine allows the elimination of panel-like structures. However, this method is affected by objects that cause the panel geometry to be lost, as well as edge distortion. In the case of the proposed method with deep learning, the process of image normalization and training with data augmentation allows to overcome the new method. However, it fails in prediction due to loss of edges, diffuse edges and loss of panel geometry. The errors presented by each of the methods were overcome with the post-processing step, in which distorted predictions were eliminated, and the location of panels was inferred by taking prior information into account.

The value of the accuracy metrics in Table 2 and Table 3 for the post-processing step have a value greater than 0.97 which implies that the panel detection methods are reliable, confirmed by the precision value being greater than 0.996. Finally the value of $F_{1}$ score is similar for both methods with a value of 0.98 for all detections.

However, the reduction of false negatives by the post-processing step is greater in the case of deep learning. This is because it has a better geometry in the detection. In the new method, the greatest influence was in the reduction of false positives. Therefore the method of deep learning with the post-processing step of panels has greater feasibility of improving the number of panels detected in environments with the presence of greater amounts of anomalies.

## 5. Conclusions

The identification of solar panels is difficult with complex backgrounds especially when there are power lines parallel to the panel edges and when there are shadows of weeds on the panel edges. Nevertheless, the proposed methods for panel detection obtain a high precision in detecting the solar panels in these circumstances.

Two panel detection methods were evaluated on 100 thermal images from 11 drone flights at three solar plants. The first method involved image correction, image segmentation and classification of these segments using support vector machines trained with an optimized vector of texture descriptors, and a post-processing. This method obtained the following values of accuracy metrics in experiments: precision of 0.997, recall of 0.970 and $F_{1}$ score of 0.983. The second method is based on deep learning and a post-processing step, for which five groups of data were defined, in which 80 photos were used for training and 20 for testing, so that when all the test data were added up, the following values of accuracy metrics were obtained: precision of 0.996, recall of 0.981 and $F_{1}$ score of 0.989.

By comparing the false positives of the two methods, the post-processing step was more effective for the deep learning method, reducing them by more than 60%, demonstrating that this method allows for improved panel detection. In this paper, it is demonstrated that the two panel detection methods with a post-processing step are effective in complex backgrounds.

Future work involves the correction of the lens distortions present in the thermal images, the use of different methods for the projection of panels to locate the panels that have not been detected, the optimization of the method for use in orthomosaics, the incorporation of a layer of geographic information for the location of power lines, and the combination of the panel detection with algorithms for the detection of panel failures with their correct classification.

## Figures and Tables

**Figure 1 sensors-20-06219-f001:**
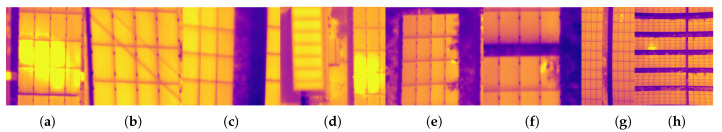
Examples of problems in thermal imaging acquired with drones with complex backgrounds. (**a**) Sunlight reflection. (**b**) Energy lines. (**c**) Poorly defined panel edges. (**d**) Like panel structures. (**e**) weeds shades. (**f**) Edges distortion by hot spots. (**g**) Lens distortion. (**h**) Display of up to 350 panels per image.

**Figure 2 sensors-20-06219-f002:**
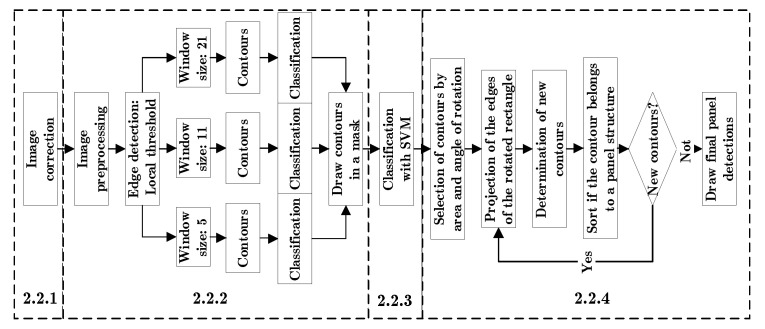
Method based on classical techniques and the numbers of the sections where the algorithm is described.

**Figure 3 sensors-20-06219-f003:**
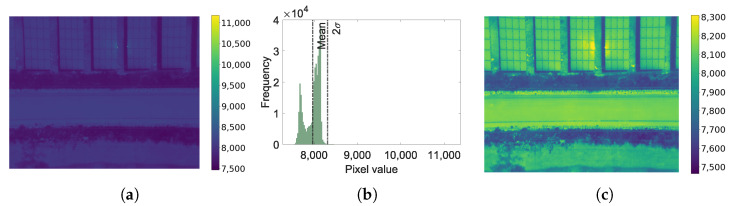
Image correction to reduce the effect of hot spots or sun reflection: (**a**) the original image; (**b**) the histogram with the probability distribution function and the values of the mean and twice the standard deviation; (**c**) the corrected image.

**Figure 4 sensors-20-06219-f004:**
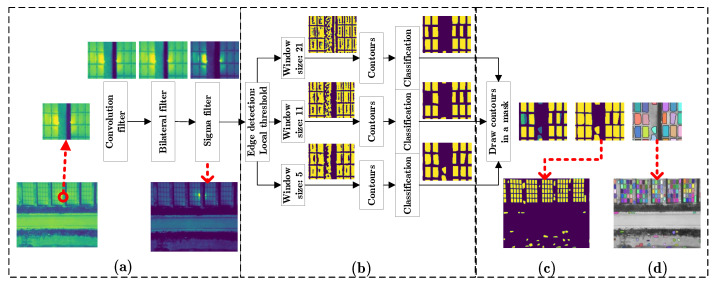
The image segmentation. (**a**) Image reprocessing. (**b**) Edge detection. (**c**) Mask with segments. (**d**) Panel detection of the first step.

**Figure 5 sensors-20-06219-f005:**
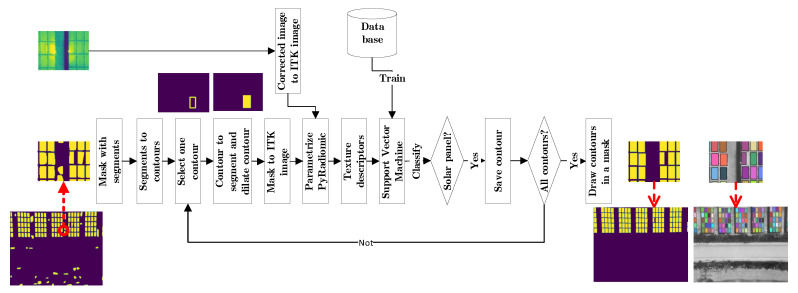
Panel classification with support vector machine.

**Figure 6 sensors-20-06219-f006:**
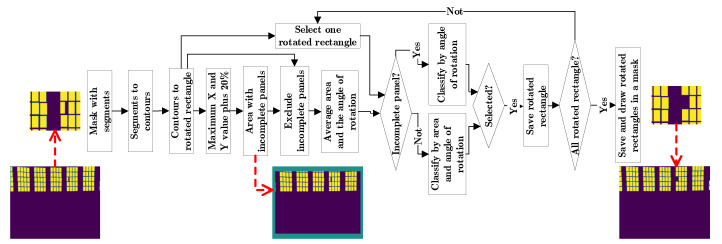
Selection of the contours of the detected panels, one panel is missing.

**Figure 7 sensors-20-06219-f007:**
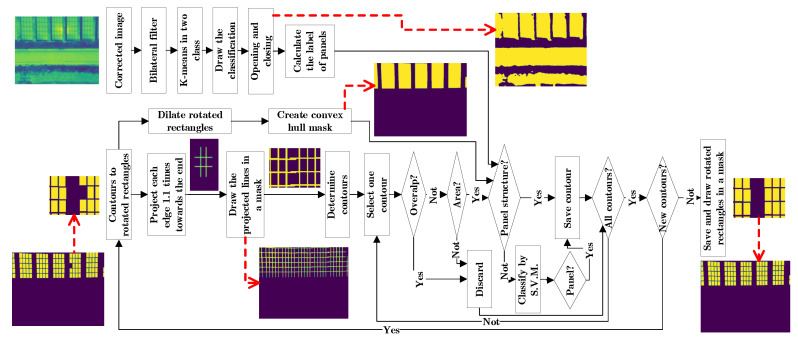
Projection of the edges of the rotated rectangle, determination of new contours and their classification.

**Figure 8 sensors-20-06219-f008:**
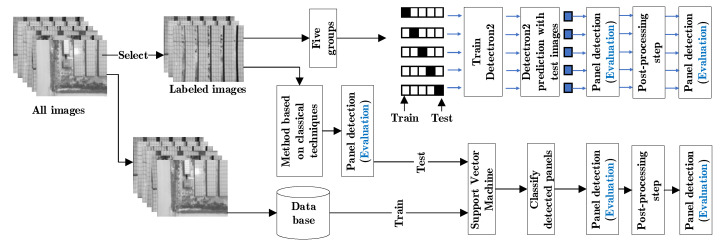
Origin of the Images to train and test for each method and the steps in which the prediction of panels is evaluated.

**Figure 9 sensors-20-06219-f009:**
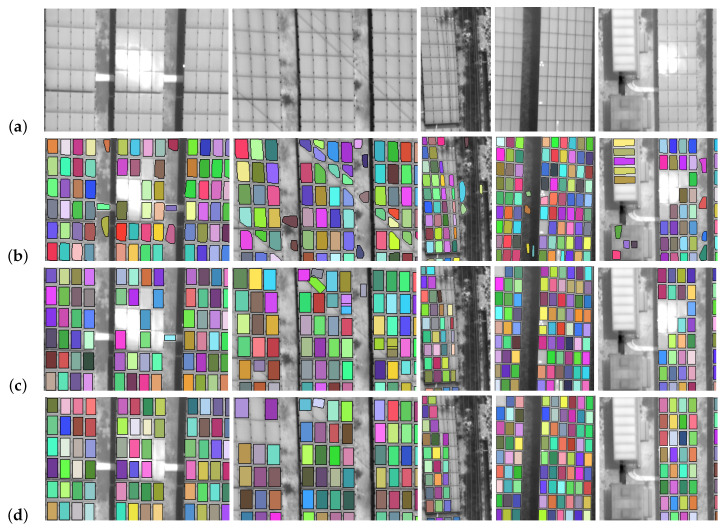
Detection of panels in thermal images with a novel method based on classical techniques and a post-processing step. Different cases of true positives, false positives and false negatives: (**a**) Thermal image in gray scale, (**b**) classification by classical segmentation with classical method, (**c**) classification with a support vector machine and (**d**) classification after post-processing.

**Figure 10 sensors-20-06219-f010:**
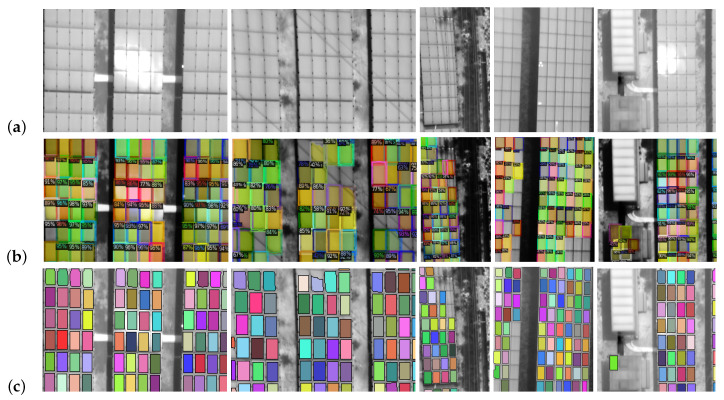
Detection of panels in thermal images with our deep learning method and a post-processing step. Different cases of true positives, false positives and false negatives: (**a**) thermal image in gray scale, (**b**) panel detection with deep learning, (**c**) after post-processing.

**Table 1 sensors-20-06219-t001:** Comparison between the proposed method and other research identifying solar panels using only thermal images with a complex background.

Method	Description	Weaknesses
Dhimish et al. [34]	Identifies hot spots in an RGB image with a color ramp.	Does not identify the affected solar panel. The thermal images have only a panel in high resolution.
Libra et al. [35]	Identifies hot spots in an RGB image with a color ramp.	Does not identify the affected solar panel. Does not segment the image
Liao et al. [31]	Identifies hot spots in an RGB image with a color ramp. Classic method based on filters for a binary classification of the image between faulty and non-faulty areas.	Does not identify the affected solar panel. Assumes that in all the photo there are only panels Lacks of methods to classify segments.
Alsafasfeh et al. [26]	Segmentation based on hot pixels detection. Classic method based on Canny edge detection	Does not identify the affected solar panel. Lacks of methods to classify segments.
Addabbo et al. [32]	Panel detection with classic methods based on template matching using normalized cross-correlation Tested on large dataset	The thermal images present a few panels in high resolution. It does not report or present solutions for the feautures 1, 4, and 5 of the complex background.
Alfaro-Mejía et al. [28]	Classic method based on two techniques. Performs an image transformation to orthogonize the detected panel	The thermal images present a few panels in high resolution. It does not present solutions for any of the complex backgrounds
Uma et al. [33]	Classic method with segmentation of the image using the k-means clustering algorithm.	k-means is an unsupervised classification. It does not present solutions for the feautures 3, 4, and 5 of the complex background.
Zhu et al. [30]	learning with an algorithm based on a fully convolutional neural network and a dense conditional random field	It does not propose a solution to identify panels that remained undetected by the deep learning method.
Greco et al. [29]	Deep learning with a convolutional neural network framework called ’You only Look Once’ (YOLO) Tested on large dataset	It does not propose a solution to identify panels that remained undetected by the deep learning method.

**Table 2 sensors-20-06219-t002:** The precision, recall and $F_{1}$ score metrics for panel detection in thermal images for three steps of our novel method based on classical techniques: segmentation, classification with a support vector machine (SVM) and after post-processing.

Method	T Positives	F Positives	F Negatives	Precision	Recall	$F_{1}$ Score
Segmentation	17,537	2262	707	0.886	0.961	0.922
Segmentation + SVM	17,599	274	645	0.985	0.965	0.975
Segmentation + SVM + post-processing	17,688	60	556	0.997	0.970	0.983

**Table 3 sensors-20-06219-t003:** The precision, recall and $F_{1}$ score metrics for panel detection in thermal images using deep learning method, before and after post-processing.

Method	Group	T Positives	F Positives	F Negatives	Precision	Recall	$F_{1}$ Score
Deep learning	1	3400	16	138	0.995	0.961	0.978
	2	3848	6	267	0.998	0.935	0.966
	3	3117	25	199	0.992	0.940	0.965
	4	3615	28	170	0.992	0.955	0.973
	5	3342	13	148	0.996	0.958	0.976
	Total	17,322	88	922	0.995	0.949	0.972
Deep learning + post-processing	1	3468	15	70	0.996	0.980	0.988
	2	4050	6	65	0.999	0.984	0.991
	3	3246	17	70	0.995	0.979	0.987
	4	3694	17	91	0.995	0.976	0.986
	5	3443	8	47	0.998	0.987	0.992
	Total	17,901	63	343	0.996	0.981	0.989

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
