# Peer review of "Solar Panel Detection within Complex Backgrounds Using Thermal Images Acquired by UAVs"

_sensors, 2020, doi:10.3390/s20216219_

Round 1
Reviewer 1 Report
- The issue is topical, failures of PV power plants occur and fault detection is important. The drone monitoring method is appropriate. Similar issues have been addressed several times. But even so, the article brings something new.
- In addition to the infrared camera, faults can also be diagnosed by data monitoring from the power plant. This should be discussed in the introduction or in the discussion of the results (see for example the papers:
"Yahyaoui, I., Segatto, M.E.V. A practical technique for on-line monitoring of a photovoltaic plant connected to a single-phase grid. Energy Conversion and Management, 2017, 132, 198–206, DOI:10.1016/j.enconman.2016.11.031."
"Beránek, V., Olšan, T., Libra, M., Poulek, V., Sedláček, J., Dang, M-Q., Tyukhov, I.I., New Monitoring System for Photovoltaic Power Plants’ Management. Energies, 2018, 11, 10, Article No. 2495, doi:10.3390/en11102495."
- When the power drops in some parts of the power plant, it is a sign of a fault.
- I think the following article should be mentioned in references, which talks about a similar issue:
"Libra, M., Daneček, M., Lešetický, J., Poulek, V., Sedláček, J., Beránek, V. Monitoring of Defects of a Photovoltaic Power Plant Using a Drone. Energies, 2019, 12, 5, Article No. 795, doi:10.3390/en12050795."
"Dhimish, M., Alrashidi, A. Photovoltaic Degradation Rate Affected by Different Weather Conditions: A Case Study Based on PV Systems in the UK and Australia. Electronics, 2020, 9, 4, Article No. 650, DOI:10.3390/electronics9040650."
- The conclusion is too brief.
Author Response
Dear Reviewer,
We appreciate the comments and suggestions because they allow us to tremendously improve the manuscript. Therefore, we have adapted the manuscript to these comments. Also, a revision of the writing style was made. Since the revision involved changes in the structure, the revised document is attached in which the changes are indicated in red.
The detailed answers to the comments are indicated in green below:
Comments and Suggestions for Authors
- The issue is topical, failures of PV power plants occur and fault detection is important. The drone monitoring method is appropriate. Similar issues have been addressed several times. But even so, the article brings something new.
- In addition to the infrared camera, faults can also be diagnosed by data monitoring from the power plant. This should be discussed in the introduction or in the discussion of the results (see for example the papers:
"Yahyaoui, I., Segatto, M.E.V. A practical technique for on-line monitoring of a photovoltaic plant connected to a single-phase grid. Energy Conversion and Management, 2017, 132, 198–206, DOI:10.1016/j.enconman.2016.11.031."
"Beránek, V., Olšan, T., Libra, M., Poulek, V., Sedláček, J., Dang, M-Q., Tyukhov, I.I., New Monitoring System for Photovoltaic Power Plants’ Management. Energies, 2018, 11, 10, Article No. 2495, doi:10.3390/en11102495."
- When the power drops in some parts of the power plant, it is a sign of a fault.
- I think the following article should be mentioned in references, which talks about a similar issue:
"Libra, M., Daneček, M., Lešetický, J., Poulek, V., Sedláček, J., Beránek, V. Monitoring of Defects of a Photovoltaic Power Plant Using a Drone. Energies, 2019, 12, 5, Article No. 795, doi:10.3390/en12050795."
"Dhimish, M., Alrashidi, A. Photovoltaic Degradation Rate Affected by Different Weather Conditions: A Case Study Based on PV Systems in the UK and Australia. Electronics, 2020, 9, 4, Article No. 650, DOI:10.3390/electronics9040650."
R/ We included the references in the discussion in the introduction section.
- The conclusion is too brief.
R/ We have extended the conclusion to include which anomalies are more difficult to overcome and other areas of future work.
[…] The identification of solar panels is difficult with complex backgrounds especially when there are power lines parallel to the panel edges and when there are shadows of weeds on the panel edges. Nevertheless, the proposed methods for panel detection obtain a high precision in detecting the solar panels in these circumstances. […] Future work involves the correction of the lens distortions present in the thermal images, the use of different methods for the projection of panels to locate the panels that have not been detected, the optimization of the method for use in orthomosaics, the incorporation of a layer of geographic information for the location of power lines, and the combination of the panel detection with algorithms for the detection of panel failures with their correct classification.

Reviewer 2 Report
The paper deals with the identification of solar panels by using remotely sensed data (thermal images) from UAVs.
The topic is of interest but the manuscript requires major revisions before being considered for publication.
Here the comments:
1) English editing is required.
2) Please, avoid using "We", the passive form is advised.
3) Page 1, lines 9-14: the main difference between the two methods is not well explained.
4) Figure 1 should also report false-color images to improve readability.
5) Page 2, line 54: please, improve the definition.
6) Page 2, lines 56-61: please, improve the explanation of the manuscript objective. What do authors mean by "automatic anomaly detection"? Real-time identification of hot spots is already available on the market.
7) Page 2, lines 62-68: authors refer to one method while, in the Abstract section, two methods have been claimed to be developed.
8) Section 2.1: camera focal length, flight height, UAV speed, images overlapping percentage (frontal and side), time of day at which flights were done, ground control points adopted, emission values, and ground sample distance are missing.
9) Figure 3: why not using the median?
10) Section 2.2.2: Figure 4 is not sufficient to explain all the steps reported in section 2.2.2. Please, identify a sub-portion of an image to also explain in a graphical way all the steps reported in section 2.2.2. This helps in the readability of lines 123-146. Indeed, the graphical representation of the algorithm steps in now missing.
11) Sections 2.2.3 and 2.2.4: the same as point 10).
12) Section 2.3.1: it is not clear what has been developed by authors.
13) Figures 6 and 7: once I got all those classified panels, how could I cluster them to extract valuable information?
Author Response
Dear Reviewer,
We appreciate the comments and suggestions because they allow us to tremendously improve the manuscript. Therefore, we have adapted the manuscript to these comments. Also, a revision of the writing style was made. Since the revision involved changes in the structure, the revised document is attached in which the changes are indicated in red.
The detailed answers to the comments are indicated in green below:
Comments and Suggestions for Authors
The paper deals with the identification of solar panels by using remotely sensed data (thermal images) from UAVs.
The topic is of interest but the manuscript requires major revisions before being considered for publication.
Here the comments:
1) English editing is required.
R/ We have revised the document and improved the English language.
2) Please, avoid using "We", the passive form is advised.
R/ We changed the sentences in the manuscript into the passive form.
3) Page 1, lines 9-14: the main difference between the two methods is not well explained.
R/ We have clarified the difference: The first method is based on edge detection and classification, in contrast to the second method which is based on training a region based convolutional neural network to identify a panel.
4) Figure 1 should also report false-color images to improve readability.
R/ We have adapted Figure 1.
5) Page 2, line 54: please, improve the definition.
R/ We have improved the definition: Edge detection fails due to image saturation in areas with hot spot or sunlight reflection
6) Page 2, lines 56-61: please, improve the explanation of the manuscript objective. What do authors mean by "automatic anomaly detection"? Real-time identification of hot spots is already available on the market.
R/ We have clarified the objective: […] because the correct classification is based on geometries in the panel or between panels. The development of methodologies for the detection of solar panels directly from the thermal image will allow the design of a fully automated anomaly classification system.
7) Page 2, lines 62-68: authors refer to one method while, in the Abstract section, two methods have been claimed to be developed.
R/
Original: We propose a novel post-processing step of "detecting/identifying entire strings or modules of solar panels" in order to predict missing (i.e. undetected) solar panels. The post-processing improves the recall of both methods a lot.
Corrected: A novel method predicting} missing (i.e. undetected) solar panels. This method is used as a post-processing step for both detection algorithms: based on classical techniques and deep learning. The post-processing improves the recall of both methods.
8) Section 2.1: camera focal length, flight height, UAV speed, images overlapping percentage (frontal and side), time of day at which flights were done, ground control points adopted, emission values, and ground sample distance are missing.
R/ The flight height are incorporated in the paper. However, the methods are optimized to only use the information from the thermal image, therefore the other information is not relevant for the research.
9) Figure 3: why not using the median?
R/ Because the standard deviation uses the mean as a reference value
10) Section 2.2.2: Figure 4 is not sufficient to explain all the steps reported in section 2.2.2. Please, identify a sub-portion of an image to also explain in a graphical way all the steps reported in section 2.2.2. This helps in the readability of lines 123-146. Indeed, the graphical representation of the algorithm steps in now missing.
R/ We have included the graphical representation of the steps of the algorithm in section 2.2.2, with detailed images of the changes made
11) Sections 2.2.3 and 2.2.4: the same as point 10).
R/ We have included the graphical representation of the steps of the algorithm in section 2.2.4 and 2.2.4, with detailed images of the changes made
12) Section 2.3.1: it is not clear what has been developed by authors.
R/ The change is in the document
The detectron2 model is trained with the procedures reported for a new data set […]
13) Figures 6 and 7: once I got all those classified panels, how could I cluster them to extract valuable information?.
R/ This is a future work with the creation of the routines required for the detection of panel failures with their correct classification.

Reviewer 3 Report
This is an interesting approach for automatically identifying solar panels in aerial thermal photographs taken by drones. Actually, two approaches are tested with apparently good results. Nevertheless, as I understood, the deep learning technic seems to demand a lot of previous labelling. The authors have not said if the labelling is made by a human operator.
As I began to read the paper, I thought the objective of the exercise would be to detect defect panels, not just panels. This would be the reason for using thermal images. Another shortcoming of the paper is that panels are detected out of its context, I mean, without geospatial reference. In a practical case, on a photovoltaic central with hundreds of panels, one would want to know where a defect panel is, not just that there are lots of panels. This fact is known already.
My question is: can the authors detect also defect panels? Can the colors of the result be part of a scale of functionality of the respective panel?
Another question: each detected rectangle is a panel or a cell? Nothing is said about the scale of the images or the flight height of the drone.
Final question: Can the procedure be applied to a thermal orthomosaic of a whole central? Or there is a limitation of application to one isolated image?
Perhaps you could address these questions in your introduction or conclusion chapters.
More comments and questions are indicated in the attached pdf file.

Author Response
Dear Reviewer,
We appreciate the comments and suggestions because they allow us to tremendously improve the manuscript. Therefore, we have adapted the manuscript to these comments. Also, a revision of the writing style was made. Since the revision involved changes in the structure, the revised document is attached in which the changes are indicated in red.
The detailed answers to the comments are indicated in green below:
Comments and Suggestions for Authors
This is an interesting approach for automatically identifying solar panels in aerial thermal photographs taken by drones. Actually, two approaches are tested with apparently good results. Nevertheless, as I understood, the deep learning technic seems to demand a lot of previous labelling. The authors have not said if the labelling is made by a human operator.
R/ corrected: Labeling is done in the visual interface by a user who eliminates false positives and corrects false negatives
As I began to read the paper, I thought the objective of the exercise would be to detect defect panels, not just panels. This would be the reason for using thermal images. Another shortcoming of the paper is that panels are detected out of its context, I mean, without geospatial reference. In a practical case, on a photovoltaic central with hundreds of panels, one would want to know where a defect panel is, not just that there are lots of panels. This fact is known already.
R/ [… ] hot spots can be catalogued at different levels of failure, which are associated with their geometry. [… ] This paper focuses on the first step for a proper diagnosis of the fault, in which a solar panel detection is necessary to delimit the region of interest for the detection and classification of the anomaly, because the correct classification is based on geometries in the panel or between panels.
My question is: can the authors detect also defect panels? Can the colors of the result be part of a scale of functionality of the respective panel?
R/ corrected: Future work involves the correction of the lens distortions present in the thermal images, the use of different methods for the projection of panels to locate the panels that have not been detected, the optimization of the method for use in orthomosaics, the incorporation of a layer of geographic information for the location of power lines, and the combination of the panel detection with algorithms for the detection of panel failures with their correct classification.
Another question: each detected rectangle is a panel or a cell? Nothing is said about the scale of the images or the flight height of the drone.
R/ corrected: […] The database contains to 11 flights from three different solar plants, with a relative flight height of 34 to 56 meters.
The focus is on the detection of panels, regardless of the height of flight of the drone, however structures smaller than 100 square pixels are discarded, because if these were panels you would have very little detail for proper diagnosis.
Final question: Can the procedure be applied to a thermal orthomosaic of a whole central? Or there is a limitation of application to one isolated image?
R/ It is important to note that the proposed method is optimized for a single image, which requires correction for hot spots, incomplete panels and lens distortion. However, this method in a simplified mode can be used in orthophotos, so this development is proposed as future work.
corrected: Future work involves the correction of the lens distortions present in the thermal images, the use of different methods for the projection of panels to locate the panels that have not been detected, the optimization of the method for use in orthomosaics, the incorporation of a layer of geographic information for the location of power lines, and the combination of the panel detection with algorithms for the detection of panel failures with their correct classification.
Perhaps you could address these questions in your introduction or conclusion chapters.
More comments and questions are indicated in the attached pdf file.
R/ The comments were corrected in the document, and are highlighted in red, however the following comments are extracted which have response in another location.
why less than four sides? Shouldn't they be rectangles?:
R/ There are also triangles due to distortions of the complex background and by panels cut out at the edge of the image.
why is this a mask? it seems to me to be a classified image. Have the colors a meaning? why are they all different?
R/ corrected: To facilitate the evaluation each segment is drawn with a different color on the grayscale corrected image
a contour is a line. It does not have an area. Please be more precise in all the geometric descriptions.
R/ corrected: the OpenCV library is used, which allows to calculate for an contour: the image area, a rotated rectangle, the angle of rotation and convert the contour to an image segment

Round 2
Reviewer 2 Report
The manuscript is now suitable for publication.
This manuscript is a resubmission of an earlier submission. The following is a list of the peer review reports and author responses from that submission.